# Decoding decision-making behavior from sparse neural spiking activity

**Yuhang Zhang** (iD) [1,2☯], **Tao Sun** [1,2,3☯], **Boyang Zang** [1,2], **Sen Wan** (iD) [1,2*]

**1** Department of Automation, Tsinghua University, Beijing, China, **2** Institute for Brain and Cognitive Sciences, Tsinghua University, Beijing, China, **3** School of Control Science and Engineering, Dalian University of Technology, Dalian, China

☯ These authors contributed equally to this work.

* wansen@tsinghua.edu.cn

**Data availability statement:** Reproducible electrophysiology dataset provided by the IBL are available at https://int-brain-lab.github.io/iblenv/notebooks_external/data_release_repro_ephys.html. All data and

## Abstract

Decoding animal decision-making behavior from neural spike data emerges as a particularly challenging problem in neuroscience. To address this, we have devised a decision-making decoding model, incorporating a channel attention bi-directional long short-term memory network (CA-BiLSTM), with the aim of effectively parsing sparse neural spike data across multiple brain regions. Notably, the attention mechanism embedded within this model serves the crucial purpose of adeptly localizing neurons integral to decision-making stability within the task at hand. Specifically, when applied to the reproducible electrophysiology dataset from the International Brain Laboratory (IBL), our proposed model has demonstrated a remarkable capacity for accurately forecasting decision-making behavior in mice. Consequently, this investigation furnishes a novel perspective for unraveling the intricacies of neural decision-making mechanisms.

## Introduction

In the expansive realm of neuroscience, the precise quantification and prediction of animal perceptual decision-making, along with the underlying neural mechanisms, represent pivotal endeavors in unraveling the enigmas of the brain [1–4]. This pursuit is widely recognized as a fundamental challenge in comprehending the intricacies of mental operations. Specifically, this field aims to explicate and forecast the macroscopic behavioral strategies displayed by animals when confronted with choices, through the analysis of measurable neural activity data and the application of sophisticated quantitative models. This process constitutes foundational steps in unveiling how the brain processes sensory inputs and translates them into decision commands. Essentially, it represents the key in deciphering the workings of the brain, namely, the transition from the microscale activities of neurons to the macroscopic manifestations of behavior [5,6].

Perceptual decision-making research necessitates a profound understanding of the activity patterns of individual neurons and neural networks, closely associating these microscale activities with the overall decision-making behavior of animals [7,8]. This approach constructs a seamless framework for understanding from neurophysiology to behavioral science. Through precise measurement and analysis of neural signals, such as action potential firing

code used for running experiments, model fitting, and plotting is available on a GitHub repository at https://github.com/ZhangYH321/CA-BiLSTM/tree/master.

**Funding:** This work was supported in part by the National Key Research and Development Project of China (2022YFF1202900 to SW), in part by the Young Elite Scientist Sponsorship Program of China Association for Science and Technology (YESS20240994AE to TS), in part by the National Natural Science Foundation of China (62401331 to SW; 62303262 to SW; 62088102 to SW; U21B2013 to SW), in part by the Beijing Nova Program (2021B00003292 to SW; 20220484216 to SW), in part by the National Key Research and Development Project of China (2023YFC2415600 to SW), and in part by the MOST (2020AA0105500 to SW; 2022YFF1202904 to SW). The funders had no role in study design, data collection and analysis, decision to publish, or preparation of the manuscript.

**Competing interests:** The authors have declared that no competing interests exist.

rates and synchronized activities of neural clusters, mathematical models can be established to capture the essential features of neurodynamics during the decision-making process and infer the decision tendencies of animals based on these models [9,10].

Early decision-making research primarily analyzed animal decision behavior by examining the spiking activity of local neural populations [11–13]. For instance, the drift-diffusion model (DDM) [14–16] enabled the modeling of reaction time data distributions in animal decision tasks, allowing the prediction of decision-making behavior through the accumulation of evidence from neural spiking activity [17]. Another study [18] utilized support vector machine (SVM) models to decode neural spike data from the posterior parietal cortex (PPC) to predict rat decision-making behavior. Although prior research has demonstrated the impact of neural signals from single brain regions on decision-making behavior [19,20], the cooperative interplay of multiple brain regions in influencing decision-making remains an area ripe for further exploration [21,22].

Recent studies have begun to investigate how the nonspecific responses of neurons across multiple brain regions during the decision-making process may affect behavior [23,24], even in brain regions previously considered solely sensory in nature [25–27]. For example, research has shown that attention enhances stimulus information in both frontoparietal and early visual regions before a decision is made, highlighting the role of attention as a fundamental mechanism underpinning decision making [28]. Building upon this foundation, we propose an innovative deep neural network approach, the CA-BiLSTM model, tailored for sparse neural spike data across multiple brain regions. BiLSTM networks, known for capturing temporal dependencies in sequential data, have been effectively applied in neuroscience to model neural dynamics [29,30]. By utilizing pre-decision multi-brain region neural spike data in 25 mice, we achieved an average predictive accuracy of 79.6% and an average Area Under Curve(AUC) of 82.8%.

Additionally, to enhance the model's capacity in decoding neural spike data, we introduced an attention mechanism. Attention mechanisms, inspired by cognitive processes, allow models to focus on relevant parts of the input data, improving performance in tasks such as neural signal decoding. In our study, the attention mechanism resulted in an approximate 6-17% increase in prediction accuracy compared to traditional machine learning and deep learning methods. More crucially, the attention mechanism adeptly selects pivotal neural units pertinent to decision-making, providing insights into the neural substrates of behavior.

These findings offer valuable insights into addressing core scientific inquiries within the field of neuroscience. The dataset employed in this study originates from the publicly available reproducible electrophysiology dataset provided by the International Brain Laboratory (IBL) [31,32].

## Methods

In this section, we delve into the neural activity changes in mice during their reaction time, focusing on the aforementioned perceptual decision-making problem. We have established an end-to-end deep learning model, the CA-BiLSTM, to facilitate a deeper understanding of their decision-making process.

**The model:** We propose an end-to-end prediction model, the CA-BiLSTM, which comprises a channel attention network structure, BiLSTM recurrent units, and fully connected layers, as illustrated in Fig 1C. The model takes neural spike signals from mouse neurons preceding decisions as input and produces output in the form of the mouse's decision-making action.

Perceptual decision-making is a collaborative effort involving multiple brain regions. To assess the varying impact of neurons in different brain regions on decision-making in mice,

we devised a channel attention structure to gauge the magnitude of influence of distinct neurons on the predictive outcomes of mouse decision-making. This structure employs the *tanh* function as an activation function and acquires attention excitation through residual connections. Its purpose is to diminish the influence of non-decision-relevant neurons on the outcomes while bolstering the influence of decision-relevant neurons. Neural activity is encoded as temporal sequences of binned spike counts. Therefore, we employed BiLSTM to capture the temporal information of neural firing during the mouse decision-making process. This BiLSTM architecture facilitates robust memory retention and comprehensive context information capture within the neural activity data by transmitting information in both forward and backward directions. Ultimately, post-processing of features is executed through layer normalization and fully connected layers to yield the probabilities of mouse decision-making action.

In the CA-BiLSTM model, the architecture and output of the channel attention mechanism are as follows:

$$\text{MLP}(X) = W_b\left(g\left(W_a X\right)\right)$$
$$E_c = \tanh\left(\text{MLP}(\text{AvgPool}(X)) + \text{MLP}(\text{MaxPool}(X))\right)$$

(1)

where $X$ represents the matrix of neural spike data. $W_a$ and $W_b$ represent the weights of the fully connected layers in the channel attention structure, and the weights for max-pooling and average-pooling are shared. The function $g(\bullet)$ denotes the activation function, set to ReLU in this paper, while $\text{AvgPool}(\bullet)$ and $\text{MaxPool}(\bullet)$ represent temporal average-pooling and max-pooling along the time dimension, respectively. The output weights $E_c \in \mathbb{R}^{1 \times N}$ have the same dimension as the number of channels.

The weighted output after the attention mechanism is then added to the original data through a residual connection, as specified by the following formula:

$$o_c = X + E_c X$$

(2)

where $o_c$ represents the output after being weighted by the attention mechanism and connected through a residual connection. Subsequently, the output is fed into the BiLSTM network, where each LSTM unit consists of an input gate $i_t$, a forget gate $f_t$, and an output gate $o_t$. The hidden units maintains internal state memory at time step $t$. The BiLSTM is a bidirectional structure, comprising both forward and backward propagation layers, taking into full consideration the interdependencies among sequential data. By combining the outputs of the forward and backward propagation layers, it effectively integrates contextual features from the sequence. In this study, the hidden layer size was set to 128, and a 2-layer BiLSTM structure was employed. The formula for the forward propagation layer is as follows:

$$\vec{f}_t = \sigma\left(\vec{W}_f\left[\vec{h}_{t-1}^{(k)}, x_t^{(k)}\right] + \vec{b}_f\right)$$
$$\vec{i}_t = \sigma\left(\vec{W}_i\left[\vec{h}_{t-1}^{(k)}, x_t^{(k)}\right] + \vec{b}_i\right)$$
$$\tilde{C}_t = \tanh\left(\vec{W}_c\left[\vec{h}_{t-1}^{(k)}, x_t^{(k)}\right] + \vec{b}_c\right)$$
$$\vec{C}_t = \vec{f}_t \circ C_{t-1} + \vec{i}_t \circ \tilde{C}_t$$
$$\vec{o}_t = \sigma\left(\vec{W}_o\left[\vec{h}_{t-1}^{(k)}, x_t^{(k)}\right] + \vec{b}_o\right)$$
$$\vec{h}_t = \vec{o}_t \circ \tanh\left(\vec{C}_t\right)$$

(3)

where $\vec{C}_t$ and $\vec{h}_t$ represent the cell state vector and output vector in the forward pass for layer $k$, $\circ$ represents element-wise multiplication, and $T$ stands for the maximum time step, which, in this context, corresponds to the number of segments. $K$ represents the maximum number of layers. The entire BiLSTM network structure generates a sequence of outputs $H = \left[\vec{h}_T^K, \overleftarrow{h}_T^K\right]$ from each recurrent unit. The outputs from both the forward and backward passes are combined to incorporate contextual features from both past and future time steps. The network's output values, denoted as $h = \{[\vec{h}_1^K, \overleftarrow{h}_1^K], [\vec{h}_2^K, \overleftarrow{h}_2^K], ..., [\vec{h}_T^K, \overleftarrow{h}_T^K]\}$, undergo layer normalization and are then fed into a fully connected layer to produce prediction probabilities. The specific formulas are as follows:

$$y = \text{sigmoid}\left(W_d\left(g\left(W_c\tilde{H} + b_c\right)\right) + b_d\right) \tag{4}$$

where $W_c$ and $W_d$ represent the weight matrices of the fully connected layer, $g(\bullet)$ denotes the activation function, which is set to *ReLU* in this study. *sigmoid*$(\bullet)$ is used to constrain the output within the $(0, 1)$ range, representing prediction probabilities.

**Loss function:** Given the presence of noise in the mouse neural data and the inherent variability in evidence accumulation during reaction time, it is important to consider the possibility of mouse lapses. While reaction times have been controlled during the experiment selection process, the neural evidence accumulation process may not be consistently engaged across all trials. In some cases, responses might be impulsive rather than based on extended evidence integration. Additionally, due to the presence of noise, there may still be some ineffective samples in the trial data. Traditional Binary Cross-Entropy (BCE) loss functions are sensitive to sample noise; therefore, this study employs a generalized cross-entropy (GCE) loss function.

$$\text{GCELoss}(x, y) = \frac{1}{n}\sum_{k=1}^{n}\frac{1 - \left(y_k x_k + (1 - y_k)(1 - x_k)\right)^q}{q} \tag{5}$$

where $y_k \in [0, 1]$, $x_k$ represents the predicted value for the $k$-th sample, while $y_k$ represents the target value for the same sample. The parameter $q$ serves as a tuning factor, and its range is within $(0,1]$. According to L'Hôpital's rule, as $q$ approaches zero, the exponential term becomes equivalent to BCE. Conversely, as $q$ approaches 1, the exponential term transforms into a form similar to Mean Absolute Error (MAE). MAE is a noise-robust loss function, but it tends to have slow training speeds and is susceptible to underfitting in classification problems. GCE strikes a balance between these two loss functions effectively, mitigating the interference from ineffective samples.

Finally, the CA-BiLSTM model is presented below in algorithmic form (Algorithm 1).

## Results

### Problem description

In this study, we leveraged the open-source multi-laboratory collaboration project initiated by the International Brain Laboratory (IBL) in 2022. This project utilized Neuropixels multi-electrode probes to record synchronized neural activity from specific brain regions of experimental mice. Data were collected from nine laboratories worldwide.

As illustrated in Fig 1A and 1B, the experimental mice were positioned in front of a screen and required to manipulate a rotating wheel to align visual stimuli presented on one side of the screen to the center. When the stimulus appeared on the left, the mice had to rotate the wheel clockwise to move the target to the center; conversely, when the stimulus appeared on the right, a counterclockwise rotation was required. The experimental design rewarded the

**Algorithm 1** CA-BiLSTM Algorithm

**Input:** Sparse spike data within the mouse's reaction time $X \in \mathbb{R}^{T \times N}$,
   training epochs $E$
**Output:** Trained CA-BiLSTM model $W_g$
1: Randomly initialize model parameters
2: $W_g^0 = \{\Theta_{attn}^0, w_{lstm}^0\}$
3: **for** $i = 0$ to $E$ **do**
4:    Calculate attention weight scores
5:    $E_c = Attn(X; \Theta_{attn}^i)$
6:    Residual connection to get mixed features $o_c$ (Eq. 2)
7:    Feed the mixed features into the BiLSTM network
8:    $H = Bi\_LSTM(o_c, w_{lstm}^i)$
9:    Calculate predicted value $\tilde{y} = \text{sigmoid}(FC(H))$
10:    Backpropagate using the GCE loss function (Eq. 5)
11: **end for**
12: **return** $W_g$

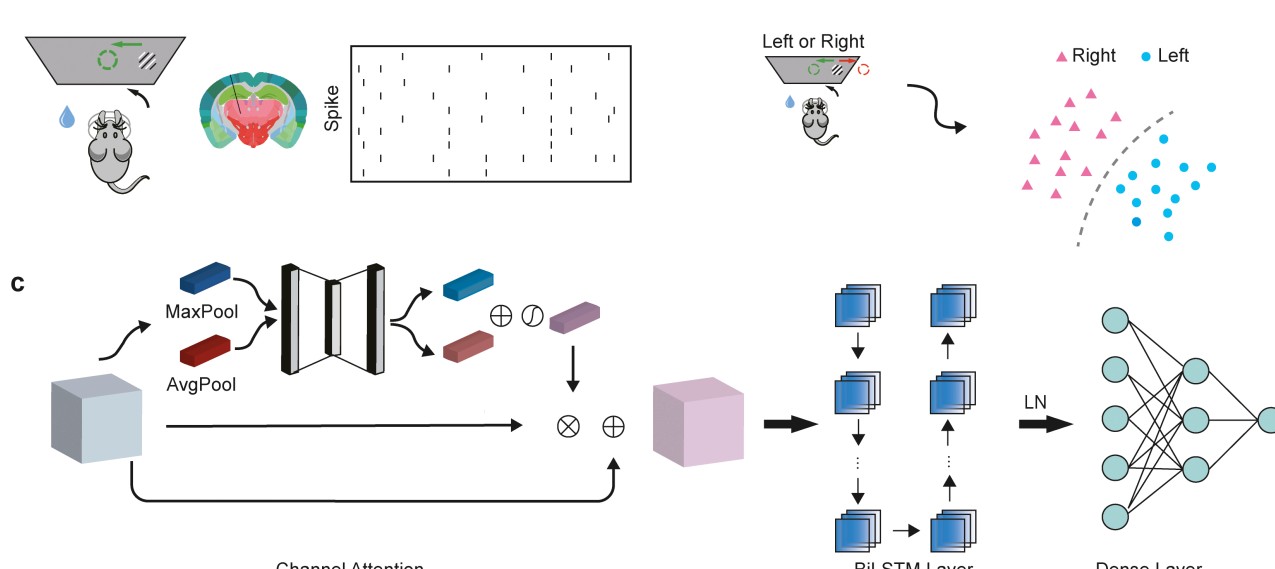

**Fig 1. |The CA-BiLSTM model accurately decodes mouse perceptual decision-making. A**, The collection of neural spike data. In the context of perceptual decision tasks in mice, we collected neural spike data pertaining to the interval between the observation of a grating and the subsequent rotation of a wheel. **B**, Predicting the decision behavior of mice (Left or right) based on neural spike data. **C**, A schematic showing the architecture of CA-BiLSTM for neural spike data. The CA-BiLSTM model consists of three components: a channel attention layer, a BiLSTM layer, and a Dense layer.

mice with water for rotating in the correct direction and to a certain angle. Incorrect rotations exceeding a predefined angle or time limit resulted in a noise punishment.

The core objective of this study is to predict the next behavioral choice of mice by analyzing their neuronal activities during the response phase. Furthermore, we aim to explore the neuronal origins involved in the decision-making process through modeling approaches.

## Experiments

In this section, we conduct a series of experiments using publicly available electrophysiology data from the IBL to validate the feasibility of the proposed algorithm.

**Dataset Description:** This study utilizes the IBL's open-access dataset, which comprises neural activity recordings obtained using Neuropixels 1.0 multi-electrode probes in freely behaving mice. The probes were stereotactically implanted at standardized coordinates (−2.0 mm AP, −2.24 mm ML, 4.0 mm DV relative to bregma) with a 15° insertion angle. Neural spiking activity was simultaneously recorded across multiple brain regions, including the primary visual cortex (VISa), dentate gyrus (DG), hippocampal CA1 field (CA1), lateral posterior thalamic nucleus (LP), and posterior thalamic nucleus (PO), while mice performed a decision-making behavioral task. Within each experimental session, the first 90 trials presented stimuli with equal probability (5:5 ratio) on the left and right sides. Subsequently, the probability distribution of stimulus presentation alternated between biased conditions of 2:8 and 8:2 (left:right) ratios. Concurrently, stimulus contrast systematically varied among six levels (100%, 25%, 12%, 6%, and 0%), allowing for the assessment of neural responses across both probabilistic and perceptual dimensions of the task.

**Data Selection Criteria:** To ensure data reliability and reproducibility, we implemented a rigorous two-stage screening process. First, we applied session-level quality control based on the IBL's standardized criteria [31]. Second, we performed trial-level filtering using the following exclusion criteria:

1. Temporal Constraints: Trials with response times outside the 30 ms–1 s range or with wheel rotation durations exceeding 1 s were discarded to maintain consistent behavioral response dynamics.
2. Behavioral Validity: Trials in which mice failed to make a choice or were presented with zero-contrast stimuli were excluded to ensure unambiguous behavioral responses and perceptual engagement.
3. Minimum Trial Count: Sessions containing fewer than 250 trials were omitted to guarantee sufficient data for robust statistical modeling.
4. Choice Balance: Sessions exhibiting strong directional bias (left/right choice proportions outside 45–55%) were excluded to prevent dataset imbalance.

These criteria were designed based on the statistical properties of murine decision-making behavior to maximize data quality and uniformity. After filtering, we retained 25 high-quality session datasets, each representing a unique mouse from different experimental cohorts (Fig 2 and Table 1).

**Data pre-processing:** The choice of neural signal preprocessing strategy has a deterministic influence on neural activity analysis. Fig 3A illustrates the comparison between two distinct preprocessing methods for neural spike signals. The upper panel presents the time-interval-based preprocessing method, which quantifies neural activity by counting spike

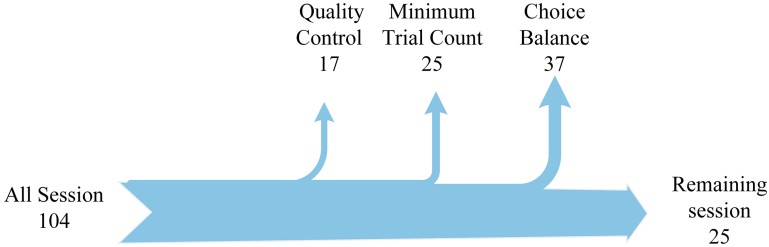

**Fig 2. |Session filtering process.**

**Table 1. Mouse information after pre-processing.**

| Animal | Initial Trials | Behavioral Validity | Temporal Constraints |
|---|---|---|---|
| UCLA005 | 772 | 678 | 537 |
| PL050 | 874 | 757 | 602 |
| MFD_05 | 620 | 538 | 281 |
| ZFM-02369 | 733 | 638 | 439 |
| UCLA015 | 610 | 537 | 369 |
| NR_0031 | 489 | 433 | 309 |
| ibl_witten_29 | 663 | 575 | 426 |
| CSHL058 | 626 | 551 | 407 |
| CSHL051 | 510 | 448 | 312 |
| DY_018 | 623 | 548 | 363 |
| DY_013 | 434 | 385 | 284 |
| NYU-45 | 505 | 454 | 255 |
| KS094 | 490 | 438 | 311 |
| SWC_052 | 658 | 577 | 313 |
| NYU-26 | 1146 | 1010 | 721 |
| NYU-37 | 414 | 375 | 286 |
| CSHL052 | 812 | 710 | 500 |
| SWC_058 | 695 | 611 | 328 |
| UCLA037 | 580 | 503 | 388 |
| ZFM-02372 | 452 | 398 | 305 |
| KS023 | 641 | 564 | 430 |
| ibl_witten_13 | 850 | 748 | 667 |
| UCLA044 | 699 | 604 | 384 |
| CSH_ZAD_019 | 1013 | 890 | 578 |
| NR_0020 | 525 | 467 | 368 |

occurrences within fixed time intervals; the lower panel demonstrates the equal-segment-based preprocessing method, which uniformly divides decision times across different trials and calculates spike counts within each segment. Given the significant variations in decision times across trials, the equal-segment-based preprocessing method enables standardized dimensional representation across different trials, effectively mitigating interference from redundant information in model discrimination.

To validate the performance differences between preprocessing methods, Fig 3B presents accuracy comparisons of both preprocessing strategies using the CA-BiLSTM model. Each data point represents the mean accuracy for an individual mouse sample, while triangular markers indicate the average accuracy across all mice. The results demonstrate that for the majority of mouse samples, the equal-segment-based preprocessing method achieved significantly higher accuracy compared to the time-interval-based approach, substantiating the efficacy of the preprocessing method proposed in this study (S1 Table).

**Parameter settings:** In this study, the model was trained for 50 iterative epochs, with an early termination criterion if the validation set's loss value did not decrease for 10 consecutive iterations. $\ell_2$-norm regularization was employed to prevent overfitting. The initial learning rate was set to 0.001, and cosine annealing was used during training to adjust the learning rate, thus preventing convergence to local optima.

For data organization, neural activity data was arranged in a structured format where each sample consisted of spike counts from N neurons across T time segments within a single trial. Data batching was implemented with a mini-batch size of 128 trials, resulting in input tensors of shape [128, T, N] for each training iteration, where T=7 represents the number of equal time segments and N varies across mice depending on the number of recorded neurons. This

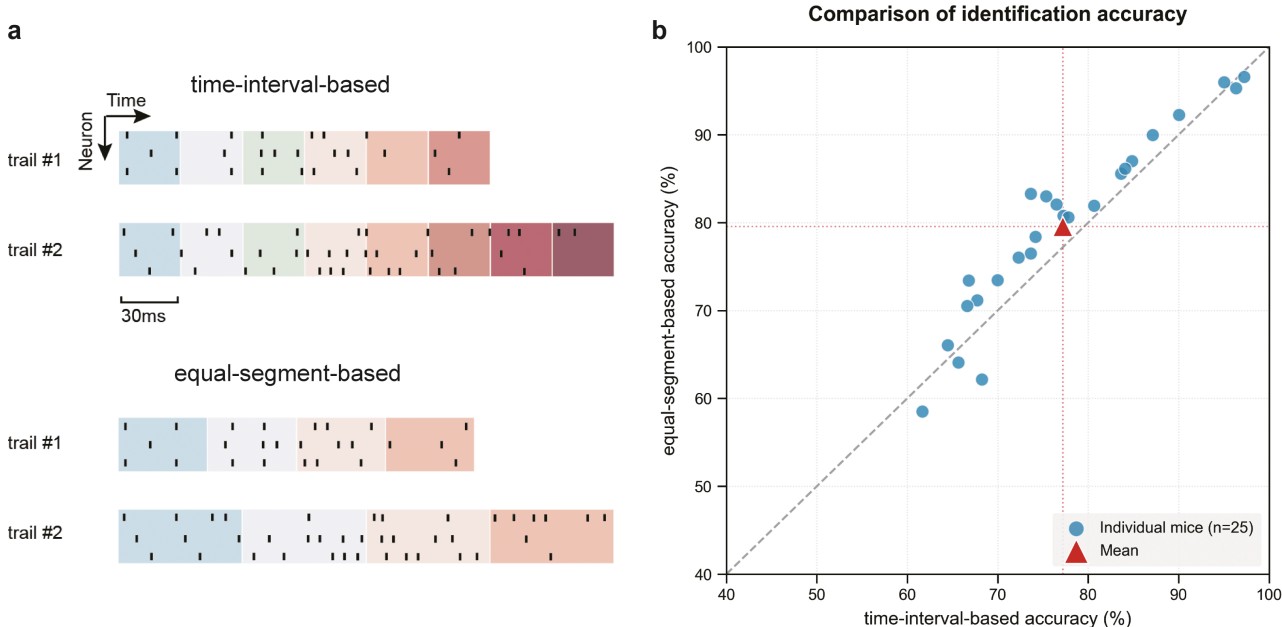

**Fig 3. |Comparison of neural signal preprocessing methods and their impact on classification performance. A,** Comparison of neural signal preprocessing strategies: The upper panel illustrates the time-interval-based preprocessing method (quantifying spike counts within fixed time windows); the lower panel shows the equal-segment-based preprocessing method (dividing decision time uniformly across trials and calculating spike counts within each segment). Due to significant variations in decision times across trials, the equal-segment-based approach enables standardized dimensional representation of data from different trials, effectively minimizing interference from redundant information. **B,** Performance comparison of preprocessing methods: Scatter plot showing classification accuracy of the CA-BiLSTM model for individual mouse samples under different preprocessing approaches, with triangular markers indicating mean accuracy across all mice. Results demonstrate that the equal-segment-based preprocessing method yields significantly higher accuracy for the majority of samples, validating the effectiveness of the proposed preprocessing strategy.

batching approach ensures that the temporal dependencies within each trial are preserved while allowing efficient parallel processing across multiple trials. Within each batch, trials were randomly sampled from the training set to prevent any potential order-related biases.

A 5-fold cross-validation approach was utilized, and the average of the highest accuracy achieved on the validation sets across the folds was considered the final model accuracy. The detailed parameters of the specific model are shown in S2 Table.

**Baseline model:** The machine learning methods employed in this study include LR [32], DT [33], KNN [34], and GLM [35] models. These methods have been widely applied in the analysis of neural data [36]. Traditional machine learning methods necessitate the vectorization of input data, resulting in some loss of temporal information inherent in the mouse decision-making process. In terms of deep learning models, we utilized CNN and MLP models. The CNN model [37] comprises one-dimensional convolutional layers with varying kernel sizes, followed by feature concatenation to predict the mouse's decision-making direction. Conversely, the MLP model [38] involves two fully connected layers for feature extraction before predicting the mouse's decision-making direction. The hidden layer sizes in these deep learning models are consistent with those in the CA-BiLSTM network.

**Result analysis:** In our comprehensive performance evaluation, we fixed the number of segments at 7 and systematically compared the average accuracy and AUC performance across 25 mouse samples under different models, with each data point representing individual mouse accuracy and AUC metrics.Fig 4A results clearly demonstrate that deep learning methods

substantially outperform traditional machine learning approaches, further confirming the significant advantages of non-linear deep learning methodologies in neural decision prediction tasks (S1 Fig). Further analysis reveals that the CA-BiLSTM model achieved average prediction accuracy and AUC of 79.6% and 82.8%, respectively, across all 25 mice, representing improvements of 0.8% and 1.1% compared to the CNN model. These findings robustly demonstrate that the CA-BiLSTM model effectively captures key neural spike features influencing mouse decision-making, while highlighting the practical value of CA-BiLSTM-extracted features for in-depth analysis of mouse decision-making behavior.

Fig 4B presents comparative accuracy and AUC metrics for different algorithms across various segment number configurations. The results clearly indicate that deep learning methods significantly outperform traditional machine learning approaches across all segment number settings, suggesting that non-linear deep learning methods possess distinct advantages in neural decision prediction domains. Notably, the CA-BiLSTM model exhibits exceptional stability across different segment number configurations, consistently outperforming other algorithmic models in both accuracy and AUC—two critical performance indicators.

While theoretically increasing bin count provides finer temporal dynamic features, this relationship is not strictly monotonic due to several competing factors. First, as bin count increases, neural spike data within each bin becomes increasingly sparse, making it difficult to capture meaningful feature information between spikes. This sparsity introduces noise and reduces the signal-to-noise ratio in each time segment, potentially degrading model performance. We observe that machine learning methods show significant accuracy decline under these conditions.

Second, our binning operation evenly distributes decision time, representing an inherent trade-off in temporal dynamic feature representation. Too few bins may obscure important temporal patterns, while too many bins may disrupt coherent neural activity patterns. Fig 4B primarily demonstrates the robustness of our CA-BiLSTM method across different bin configurations, rather than determining an absolute optimal bin count. The choice of 7 bins represents a balanced compromise based on comprehensive testing across 2-10 segments, balancing temporal resolution and data sparsity issues.

Importantly, we observed that different mice exhibited different optimal bin counts, potentially reflecting individual differences in neural processing speed and decision dynamics. The 7-bin configuration typically provides good performance across the entire population while maintaining computational efficiency. The relative stability of performance across different bin counts proves the robustness of our method, especially for the CA-BiLSTM model, which maintains higher performance than comparative methods across all bin configurations. This stability suggests that our method effectively captures critical temporal dynamics even at moderate temporal resolutions.

Our model provides crucial support for retrospective analysis of decision-related neural activity in mice, which is of paramount importance for understanding perceptual decision-making processes. Within our methodological framework, the attention mechanism [39,40] efficiently allocates differential weights to individual neurons, enabling us to precisely capture key neurons closely associated with mouse decision-making through the model's attention mechanism. Previous research has established that mouse decision-making processes involve coordinated activity across multiple brain regions, with neurons from different areas contributing to varying degrees. By adaptively adjusting channel weights, the channel attention mechanism effectively determines the model's focus on specific brain regions during prediction tasks. Accordingly, this study quantitatively evaluates the importance of different neurons using channel attention weights.

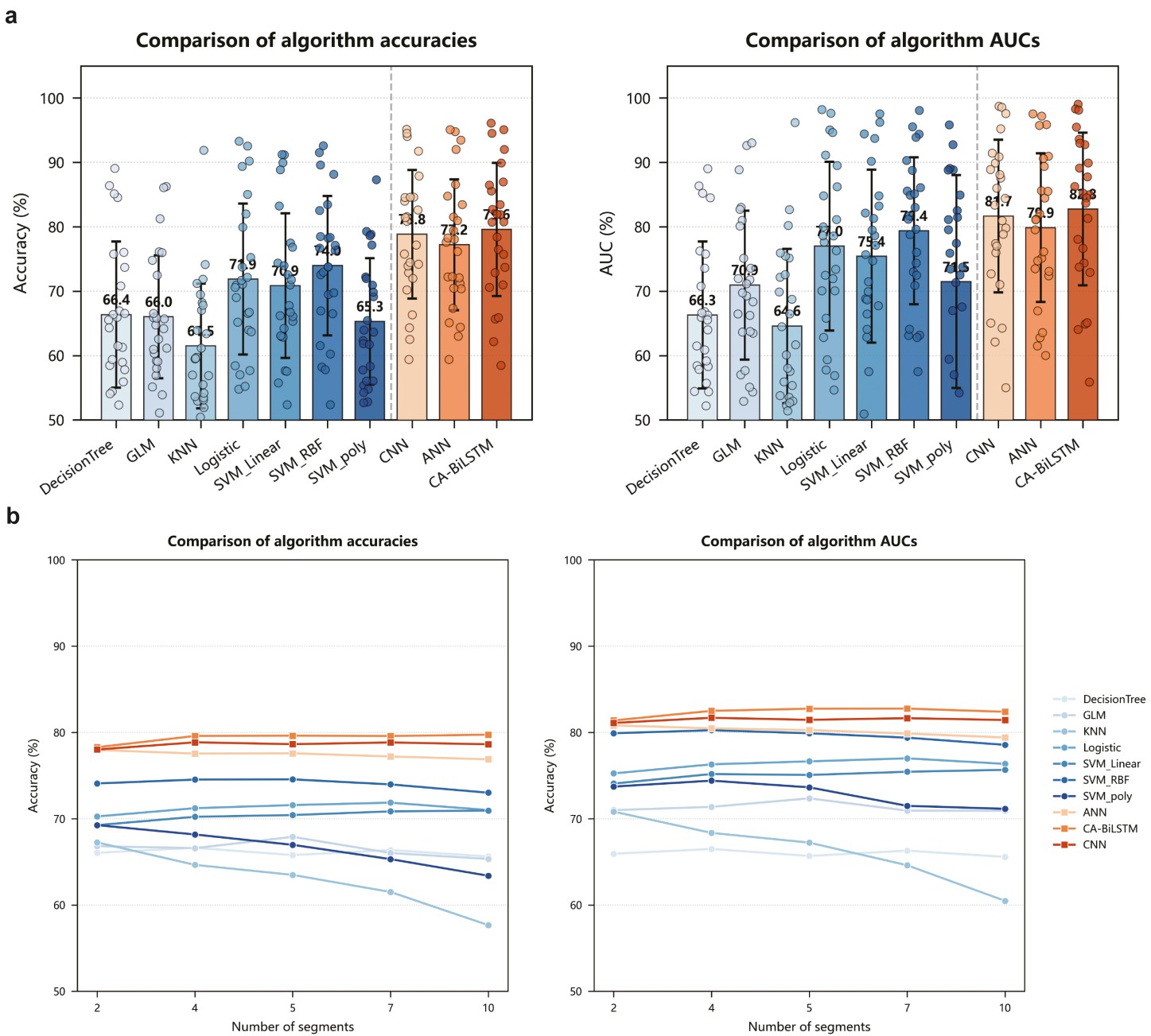

**Fig 4. |Performance comparison of different models for neural decision prediction. A.** Bar chart comparing average accuracy and AUC metrics across different models for 25 mouse samples with 7 segments. Among them, the error bars represent the standard deviation of the accuracy rates across different mice. **B.** Comparative analysis of accuracy and AUC across different segment numbers for various algorithms.

As illustrated in Fig 5, analysis of the validation set across five-fold cross-validation reveals highly consistent attention weight distributions for the same neurons across different cross-validation folds, providing compelling evidence for stable and robust associations between relevant brain regions and perceptual decision-making processes in mice. The magnitude of

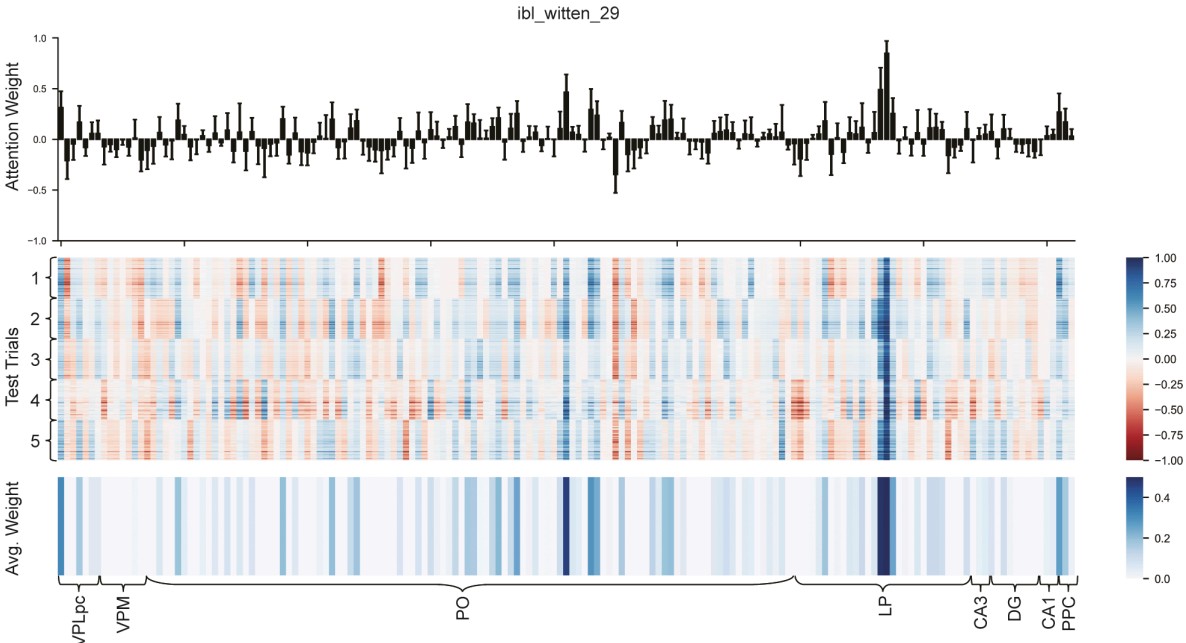

**Fig 5. |Neuronal attention weight distribution in mouse decision-making.** Visualization of attention mechanism outputs from the CA-BiLSTM model for the ibl_witten_29 mouse across five-fold cross-validation. **Top:** Distribution of attention weights for all neurons, with values from -1 to 1 indicating each neuron's contribution to the decision-making process. **Middle:** Heatmap showing trial-specific attention weights for each neuron across the test set. **Bottom:** Mean positive attention weights across all test trials. The consistent weight distributions across five-fold cross-validation demonstrate the stability and reliability of the identified neural correlates.

these weights directly reflects each neuron's influence on the mouse's perceptual decision-making process. The upper section of Fig 5 intuitively presents channel attention scores for the ibl_witten_29 mouse, quantitatively characterized through CA-BiLSTM channel attention weights, with values ranging from -1 to 1—higher values indicating stronger decision-related activity. This visualization clearly demonstrates each neuron's relative importance in the decision-making process (S2 Fig).

Building upon this attention mechanism analysis, we further investigated performance trends in mouse decision prediction after attention score-based neuronal filtering, as shown in Fig 6. Fig 6A illustrates the brain region distribution of all neurons versus filtered neurons for the ibl_witten_29 mouse, indicating that perceptual decision-making is predominantly influenced by a select subset of critical neurons (S3 Fig). To validate this finding, we systematically compared Support Vector Machine with Radial Basis Function Kernel(SVM_RBF) model performance under different selection thresholds, which determined whether neurons were retained or excluded based on their attention scores—neurons with attention scores exceeding the threshold were retained, while others were excluded. As demonstrated in Fig 6, under all selection threshold conditions, the SVM_RBF model consistently achieved superior average accuracy when using neurons filtered through the CA-BiLSTM model's attention mechanism compared to using all neurons or randomly selected neurons, robustly validating the effectiveness of the attention mechanism for neuronal selection.

Further analysis revealed that with the selection threshold set at 0.1, as shown in Fig 6C, neurons selected by the CA-BiLSTM model outperformed randomly selected neurons in the majority of individual mouse samples, highlighting the significant advantages of the attention mechanism in neuronal filtering. Notably, model accuracy was actually lower when

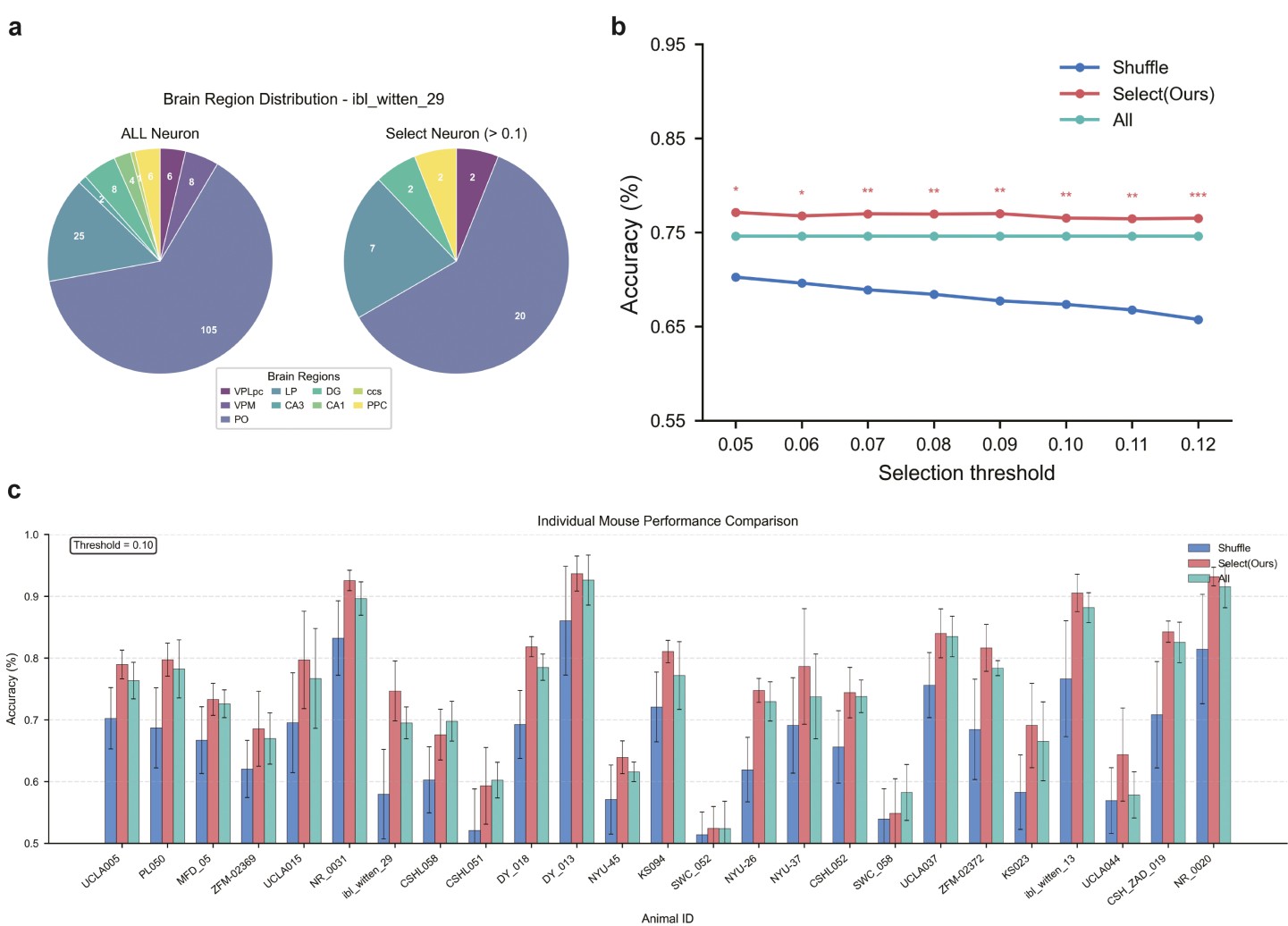

**Fig 6. |Attention-based neuronal filtering enhances decision prediction performance.** Error bars represent standard deviations; asterisks indicate significance levels (*P < 0.05, **P < 0.01, ***P < 0.001, two-tailed t-test). **A.** Brain region distribution comparison between all neurons and attention-filtered neurons for the ibl_witten_29 mouse, demonstrating that decision-making processes are driven by a specific subset of neurons. **B.** SVM_RBF model performance across different selection thresholds comparing three conditions: CA-BiLSTM attention-filtered neurons, randomly selected neurons, and all neurons. Attention-filtered neurons consistently yield superior prediction accuracy across all threshold values. **C.** Comparative performance analysis at 0.1 selection threshold across individual mouse samples. showing that CA-BiLSTM-selected neurons outperform randomly selected neurons and full neuronal ensembles in most cases, indicating that excluding irrelevant neural signals improves decision prediction accuracy. Among them, error bars represent the standard deviation of five fold cross validation.

inputs included all neuronal data compared to using only filtered neurons, confirming that substantial irrelevant neuronal information can interfere with accurate prediction of mouse decisions.

To comprehensively evaluate the effectiveness of attention-filtered neurons, we conducted in-depth analyses as presented in Fig 7. Fig 7A and 7B present validation results comparing different neuronal selection methods. We employed multiple approaches for neuronal selection—CA-BiLSTM, L1-regularized logistic regression (L1-LogReg) [41], L1-regularized multi-layer perceptron (L1-MLP) [42], analysis of variance (ANOVA) [43], and recursive feature elimination (RFE) [44]—and validated selection efficacy using SVM_RBF [45] (Fig 7A)

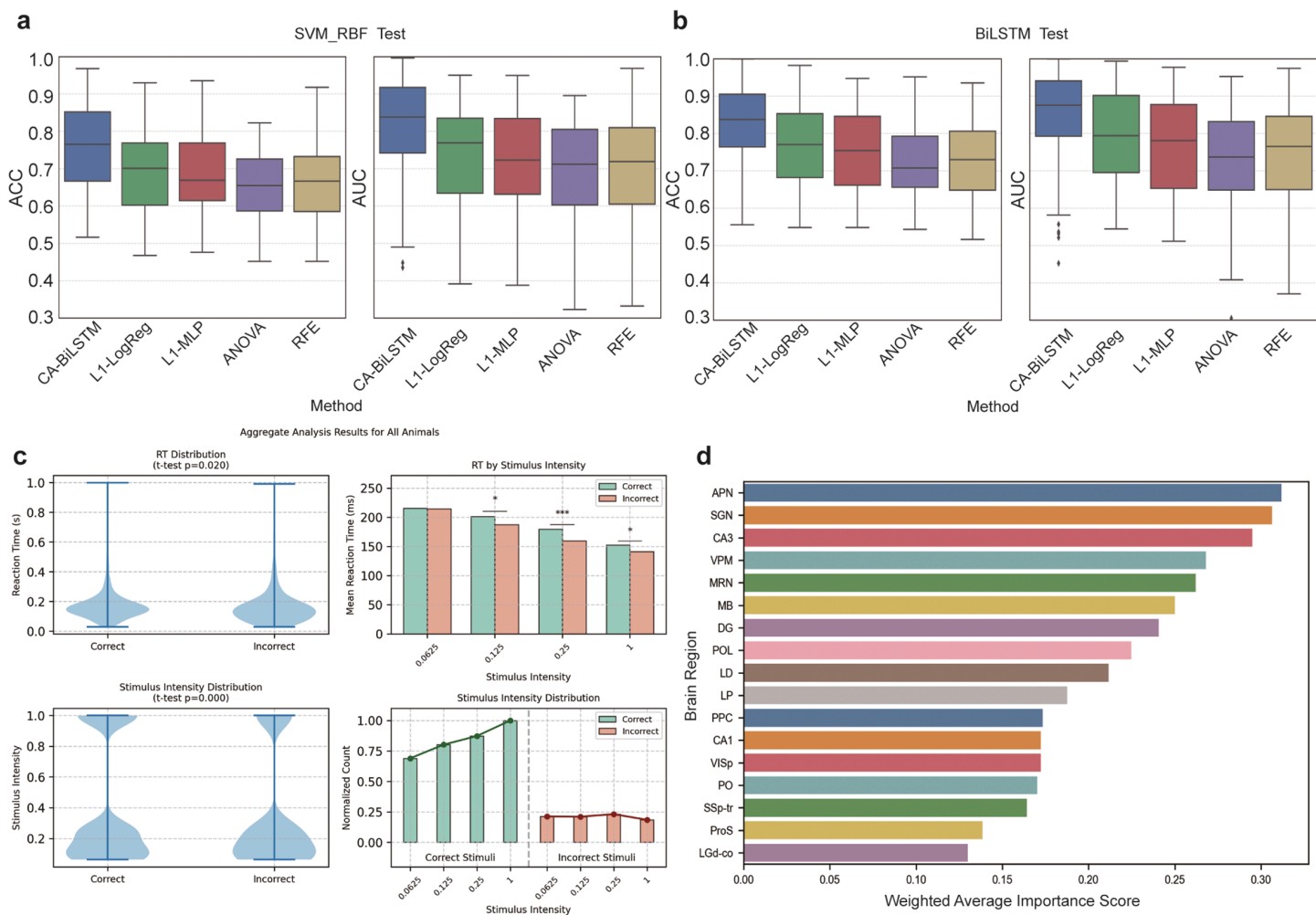

**Fig 7. |Analysis of attention-filtered neurons for decision prediction. A,B.** Comparative validation of different neuronal selection methods using (A) SVM_RBF and (B) BiLSTM models, with error bars representing standard deviations, showing CA-BiLSTM's superior performance. **C.** Comparison of mouse reaction times and stimulus intensities between trials correctly predicted by our model versus incorrectly predicted trials, revealing that trials our model predicted correctly exhibited slightly longer mouse reaction times and were associated with higher stimulus intensities. The bottom right panel shows normalized counts (divided by the maximum count across all stimulus intensities) to enable visual comparison. **D.** Average attention scores across brain regions for all mice, highlighting the importance of DG, LP, and PPC regions in decision-making processes.

and BiLSTM models (Fig 7B) as independent verification tools. The L1-LogReg method utilizes sparse regularization to automatically select features by compressing irrelevant feature weights to zero. L1-MLP extends this feature selection approach to nonlinear contexts through a single-hidden-layer perceptron with L1 penalty applied to input layer weights. ANOVA employs statistical significance-based filtering by computing F-statistics between individual neurons and output categories. RFE provides a model-based feature selection strategy through iterative training and removal of least important features using SVM classifiers.

Importantly, these analyses serve as validation of our attention weights rather than direct evidence of our method's superiority over alternatives. The SVM_RBF classifier was selected as a validation tool because it lacks temporal modeling capabilities—unlike our BiLSTM architecture—thereby providing an independent means to verify whether selected neurons contain decision-relevant information rather than being chosen due to model-specific

architectural preferences. The inclusion of BiLSTM as a secondary validation tool (Fig 7B) further confirms that attention-selected neurons perform consistently across different model architectures. Results demonstrate that neurons filtered by the CA-BiLSTM attention mechanism outperformed those selected by other methods across all validation conditions, based on both accuracy and AUC metrics. This suggests that our attention mechanism captures more meaningful neural representations related to decision-making processes, rather than simply identifying arbitrary feature subsets that enhance model performance. These findings validate the effectiveness of our attention-based neuronal selection approach.

At the neuroethological level, Fig 7C presents a detailed analysis comparing mouse reaction times and stimulus intensities between trials that our model predicted correctly versus incorrectly. When examining reaction times across different stimulus contrasts, we observed that trials our model predicted correctly exhibited slightly longer mouse reaction times than trials predicted incorrectly at most contrast levels, with an average overall difference of 16 milliseconds (T-test p-value = 0.02). This pattern was particularly pronounced at intermediate contrast levels (e.g., 0.25), whereas the difference diminished at very low contrast (0.0625). Overall, reaction times for decisions were significantly shorter at higher contrast compared to lower contrast, suggesting that mice engaged in deliberative decision-making processes. Regarding stimulus intensity, higher contrast stimuli were associated with a greater proportion of trials correctly predicted by the model, with the fewest prediction errors observed at the highest contrast (1.0). This distribution aligns with expected behavioral outcomes, as higher contrast stimuli typically enable more accurate judgments by mice. These findings suggest that longer reaction times during decision-making may reflect more stable and information-rich neural activity patterns, providing our decoding model with enhanced interpretable signals.

Finally, Fig 7D displays average attention scores across different brain regions for all mice. Among the five brain regions of primary focus in the IBL dataset (CA1, PO, DG, LP and PPC), the DG, LP, CA1 and PPC regions demonstrated notable importance in the decision-making process [46–48], consistent with existing research findings. Interestingly, while these regions show moderate average attention scores in our analysis, some regions that were not primary targets of the IBL recordings (such as the Anterior Pretectal Nucleus (APN), Suprageniculate Nucleus (SGN), and Ventral Posteromedial Nucleus of the thalamus (VPM)) exhibited higher average attention scores. This apparent discrepancy warrants explanation: the primary focus regions (CA1, PO, DG, LP, PPC) were consistently recorded across nearly all mice and typically included larger populations of neurons, many of which had lower attention weights, thus reducing their average scores. In contrast, the non-primary regions were recorded in only a subset of mice and represented by fewer neurons, some of which happened to have very high attention weights, resulting in elevated average scores.

For analytical rigor and to focus on the most reliable findings, we prioritized our analysis on the five primary brain regions targeted by the IBL study, as these regions had sufficient neuronal sampling to draw meaningful conclusions. The high attention scores observed in some non-primary regions, while potentially interesting, should be interpreted with caution due to limited sample sizes and potential sampling biases. These preliminary observations regarding non-primary regions warrant further investigation in future studies specifically designed to examine their role in visual decision-making processes.

Fig 8 illustrates distinct patterns of behavioral performance and neural decoding under varying stimulus probability conditions (P(Left) = 0.2, 0.5, 0.8). Notably, response times (RTs) reveal a marked divergence across performance metrics. As shown in Fig 8A and 8B, when assessing the accuracy of mouse behavior, we observed contrasting temporal patterns: trials with correct perceptual judgments exhibited shorter RTs compared to incorrect judgments,

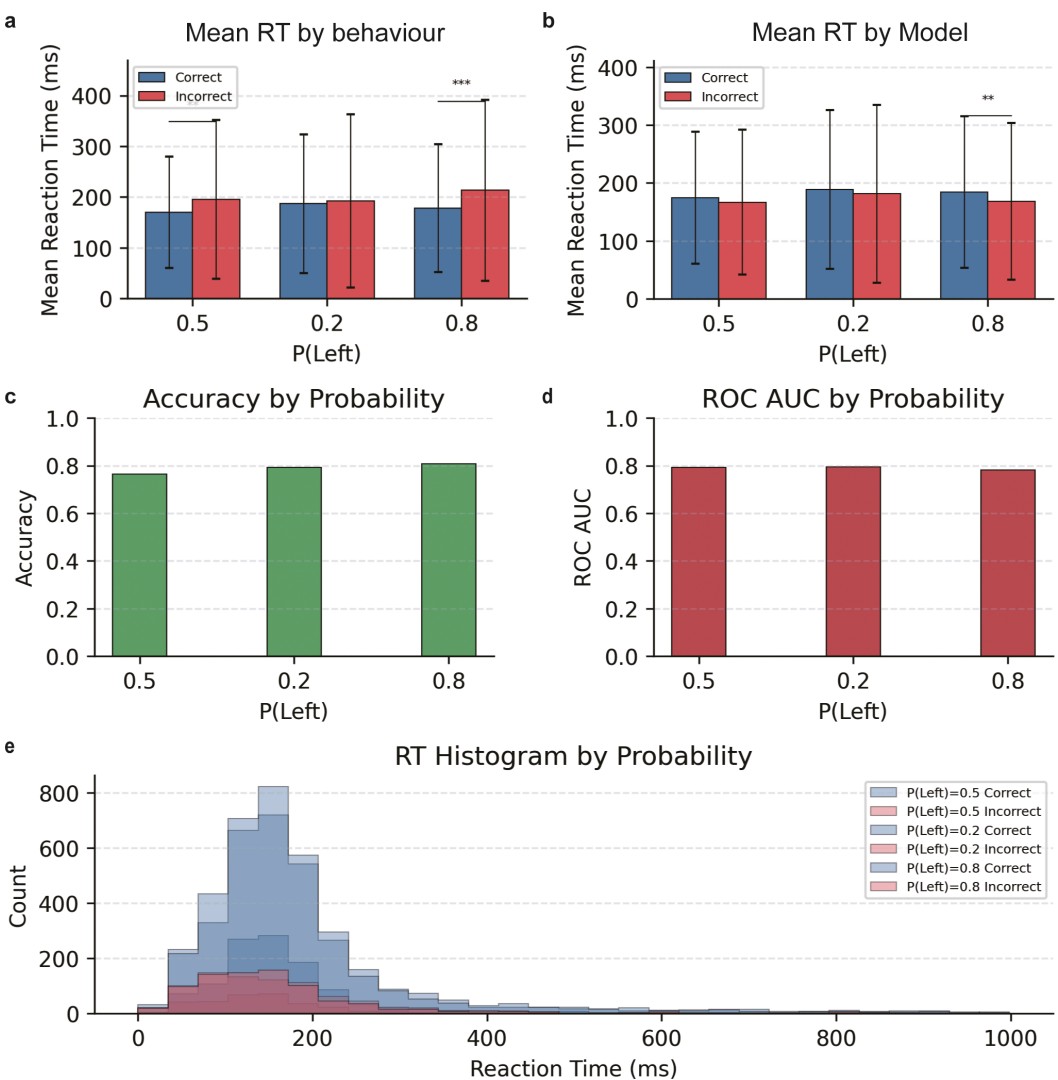

**Fig 8. |Behavioral performance and neural decoding across probability conditions.** Analysis of reaction times, accuracy, and model performance for different stimulus probability conditions (P(Left) = 0.2, 0.5, 0.8) in a perceptual decision-making task. **A.** Mean reaction times for behaviorally correct (blue) and behaviorally incorrect (orange) trials across probability conditions. **B.** Mean reaction times for model-predicted correct and model-predicted incorrect trials by probability condition. Significant differences between trial types are marked with asterisks. **C.** Model prediction accuracy across probability conditions. **D.** Neural decoder performance (ROC AUC) for each probability condition. **E.** Reaction time histograms overlaid by model prediction outcome and probability condition, showing distributional differences between model-predicted correct and incorrect responses.

with the most pronounced difference at P(Left) = 0.8. This classic speed-accuracy trade-off suggests that clearer stimulus information enables faster and more accurate responses in mice. In contrast, trials correctly predicted by our neural decoding model consistently showed longer RTs than incorrectly predicted trials, with the largest difference again at P(Left) = 0.8. This pattern indicates that extended deliberation time yields more robust and decodable neural activity, providing richer information for accurate model predictions. The contrasting temporal dynamics between neural decodability and behavioral accuracy highlight a fundamental

distinction: while faster processing enhances behavioral outcomes in mice, slightly longer RTs facilitate more decodable neural activity.

Fig 8C demonstrates that model prediction accuracy is higher under extreme probability conditions (P(Left) = 0.2 and 0.8) compared to the balanced condition (P(Left) = 0.5). This aligns with expectations, as more predictable stimulus probabilities reduce decision uncertainty, thereby improving model accuracy. However, as shown in Fig 8D and 8E, no clear trends were observed across probability conditions, with performance remaining consistent, potentially due to limited overall data volume.

## Discussion

This study presents a novel approach for decoding decision-making behavior from neural spike data using the CA-BiLSTM model. By analyzing the reproducible electrophysiology dataset from the International Brain Laboratory, we demonstrate that neural activity preceding decision-making can effectively predict mouse behavioral choices with high accuracy. Across 25 mice, our model achieved an average prediction accuracy of 79.6% and an AUC of 82.8%, significantly outperforming traditional machine learning approaches and comparable deep learning models.

The integration of a channel attention mechanism proved crucial for both performance enhancement and interpretability. This mechanism not only improved prediction accuracy by approximately 6-17% compared to conventional methods but also enabled the identification of decision-relevant neurons across multiple brain regions. Our analysis revealed that perceptual decision-making is predominantly influenced by specific neuronal subsets, with notable contributions from neurons in the DG, LP, and PPC regions. These findings align with previous studies that have implicated these regions in decision-making processes. For instance, Najafi et al. [49] highlighted the key role of PPC neurons in perceptual decisions, showing that decision-related selectivity emerges in this region during learning. Similarly, Hwang et al. [50] showed that the LP serves as a critical hub for visually guided decisions by relaying contextual information from higher visual areas to prefrontal regions.

Our observation that model performance improves when focusing on attention-filtered neurons compared to using all recorded neurons resonates with the concept of "sparse coding" in neural systems, as proposed by Olshausen and Field [51]. This principle suggests that information in the brain is encoded by the activity of a relatively small number of neurons at any given time, which maximizes efficiency and information capacity. Our findings provide empirical support for this theoretical framework in the context of decision-making, indicating that the brain may rely on specialized neuronal ensembles rather than distributed activity across all neurons in relevant regions.

The temporal dynamics revealed by our model also offer insights into how neural information unfolds during the decision-making process. The superiority of the equal-segment-based preprocessing method over fixed time intervals suggests that neural activity patterns scale with decision time, a phenomenon previously observed in primate studies by Jazayeri and Shadlen [52]. This temporal scaling property may represent a fundamental computational principle in perceptual decision-making across species.

Despite these advances, several limitations of our approach warrant critical examination. First, while our model achieves impressive predictive accuracy, it still falls short of perfect prediction, suggesting that either some decision-relevant neural signals were not captured by the recordings or that inherent stochasticity in decision-making processes limits prediction accuracy. This ceiling effect has been acknowledged in similar decoding studies [53] and represents a fundamental challenge in neural decoding.

Second, the cross-validation approach used in our study, while methodologically sound, does not test generalization across different sessions or animals. This limits our ability to claim that the identified neural patterns represent universal mechanisms rather than idiosyncratic features of particular recording sessions. Future work should explore more rigorous cross-session and cross-animal validation approaches, as advocated by Kriegeskorte et al. [54].

Third, our attention mechanism, while effective for feature selection, provides a static view of neural importance that may not capture the dynamic nature of decision processes. Recent work by Stokes et al. [55] suggests that neural representations evolve dynamically throughout the decision process, with different neurons becoming relevant at different stages. Future models could incorporate temporal attention mechanisms to address this limitation.

Fourth, the binary nature of the decision task in our study (left vs. right) represents a simplified version of real-world decisions, which often involve multiple alternatives and complex value calculations. As highlighted by Rushworth et al. [56], value-based decision-making engages additional neural circuitry beyond what is required for simple perceptual decisions. Extending our approach to more complex decision paradigms would provide a more comprehensive understanding of neural decision mechanisms.

Finally, while our model identifies correlations between neural activity and subsequent decisions, it does not establish causal relationships. Complementary approaches involving targeted neural manipulation, such as optogenetics or pharmacological interventions, would be necessary to establish the causal role of the identified neurons in decision-making, as demonstrated in recent studies by Hanks et al. [57].

Future research directions include the development of more comprehensive brain-wide attention networks to investigate how neurons across various brain regions collectively influence decision-making. Recent advances in whole-brain imaging and high-density recording techniques, as reviewed by Jun et al. [58], provide exciting opportunities for such expansive neural decoding. Additionally, incorporating real-time neural decoding into closed-loop experimental designs, as proposed by Grosenick et al. [59], could enable novel interventional studies to test hypotheses about decision-making circuits.

The methodological advances presented in our study also have potential applications beyond basic neuroscience. Similar approaches could be adapted for brain-computer interfaces to restore movement or communication in paralyzed individuals, building upon recent successes in neural prosthetics [60]. Additionally, our method for identifying decision-relevant neurons could inform targeted therapeutic approaches for disorders involving decision-making deficits, such as addiction or obsessive-compulsive disorder.

In conclusion, our CA-BiLSTM model with channel attention provides both a powerful tool for neural decoding and meaningful insights into the neural substrates of perceptual decision-making. By bridging machine learning techniques with neuroscientific questions, we advance our understanding of how distributed neural activity gives rise to discrete behavioral choices. Future work combining these computational approaches with expanded datasets and causal manipulation techniques promises to further elucidate the neural basis of decision-making across species and contexts.

## Supporting information

**S1 Table. Comparison of accuracy and AUC for different time windows.**
(XLSX)

**S2 Table. Models' parameter settings.**
(XLSX)

**S1 Fig. Loss diagram of five-fold cross-validation for partial mice.**
(TIF)

**S2 Fig. Neuronal attention weight distribution during mouse decision-making process in partial mice.**
(TIF)

**S3 Fig. Comparison of brain region distribution between all neurons and attention-filtered neurons in partial mice.**
(TIF)

**S4 Fig. Accuracy and ROC AUC performance of CA-BiLSTM model on class-imbalanced mice.**
(TIF)

## Author contributions

**Conceptualization:** Yuhang Zhang, Sen Wan.

**Data curation:** Yuhang Zhang, Tao Sun, Boyang Zang.

**Formal analysis:** Yuhang Zhang, Tao Sun, Boyang Zang.

**Funding acquisition:** Sen Wan.

**Investigation:** Yuhang Zhang, Tao Sun, Boyang Zang.

**Methodology:** Yuhang Zhang.

**Project administration:** Sen Wan.

**Resources:** Sen Wan.

**Software:** Yuhang Zhang.

**Supervision:** Tao Sun, Sen Wan.

**Validation:** Tao Sun, Boyang Zang, Sen Wan.

**Visualization:** Yuhang Zhang, Tao Sun, Boyang Zang, Sen Wan.

**Writing – original draft:** Yuhang Zhang, Tao Sun.

**Writing – review & editing:** Yuhang Zhang, Tao Sun.

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
