## [Decision Letter · Decision Letter 0]

5 Dec 2024

PCOMPBIOL-D-24-01470

Decoding Decision-Making Behavior from Sparse Neural Spiking Activity

PLOS Computational Biology

Dear Dr. Zhang,

Thank you for submitting your manuscript to PLOS Computational Biology. First of all apologies for the longish time elapsed from the submission, your work fell in the long tail of the distribution of time required to secure editors and reviewers. After careful consideration, we feel that it has merit but does not fully meet PLOS Computational Biology's publication criteria as it currently stands, in particular regarding the quality of presentation, the level of detail given to the reader, and the positioning of your results in the state of the art. Therefore, we invite you to submit a substantially revised version of the manuscript that addresses the points raised during the review process.

Please submit your revised manuscript within 60 days Feb 04 2025 11:59PM. If you will need more time than this to complete your revisions, please reply to this message or contact the journal office at ploscompbiol@plos.org. Please include the following items when submitting your revised manuscript:

We look forward to receiving your revised manuscript.

Kind regards,

Daniele Marinazzo

Section Editor

PLOS Computational Biology

Feilim Mac Gabhann

Editor-in-Chief

PLOS Computational Biology

Jason Papin

Editor-in-Chief

PLOS Computational Biology

**Journal Requirements:**

At this stage, the following Authors/Authors require contributions: Yuhang Zhang, Tao Sun, Boyang Zang, and Sen Wan. Please ensure that the full contributions of each author are acknowledged in the "Add/Edit/Remove Authors" section of our submission form.

2) Your manuscript is missing the following sections: Discussion.  Please ensure all required sections are present and in the correct order. Make sure section heading levels are clearly indicated in the manuscript text, and limit sub-sections to 3 heading levels. An outline of the required sections can be consulted in our submission guidelines here:

5) We notice that your supplementary Figures are included in the manuscript file. Please remove them and upload them with the file type 'Supporting Information'. Please ensure that each Supporting Information file has a legend listed in the manuscript after the references list.

Potential Copyright Issues:

- Figure 1; Please confirm whether you drew the images / clip-art within the figure panels by hand. If you did not draw the images, please provide a link to the source of the images or icons and their license / terms of use; or written permission from the copyright holder to publish the images or icons under our CC BY 4.0 license. Alternatively, you may replace the images with open source alternatives. See these open source resources you may use to replace images / clip-art:

**Reviewers' comments:**

Reviewer's Responses to Questions

**Comments to the Authors:**

Reviewer #1: The study proposes a novel decision-making decoding model, the Channel Attention Bi-directional Long Short-Term Memory network (CA-BiLSTM), designed to decode sparse neural spike data from multiple brain regions and predict animal decision-making behavior. The model integrates an attention mechanism to identify neurons and brain regions critical to task performance. The application to neural spike data from the International Brain Laboratory (IBL) dataset highlights its ability to forecast decision-making behavior in mice, offering a new approach to understanding neural decision-making processes.

Strengths

1. Attention Mechanism Integration: The inclusion of a channel attention mechanism represents an innovative approach to pinpointing relevant neurons and brain regions contributing to decision-making stability. This could be an important step toward interpretable models in neuroscience.

2. Application to Reproducible Data: The use of the IBL dataset, a well-documented and reproducible resource, ensures the results are transparent and can be verified by other researchers.

3. Improved Prediction: The CA-BiLSTM model successfully demonstrates its capacity to predict decision-making behavior, validating the effectiveness of combining spike data from multiple brain regions with attention-based sequence modeling.

Main points:

1. Superiority Over Simpler Models is Unclear: While the attention mechanism is stated to improve localization of task-relevant neurons and regions, its advantage over simpler model selection techniques (e.g., feature selection or regularized regression models) is not convincingly demonstrated. The dynamic nature of the attention mechanism’s contribution to overall model performance remains vague.

2. Neglect of Fine Temporal Spike Patterns: The model’s approach to partitioning spike data appears to overlook the fine-grained temporal structure of spike trains, which may encode critical decision-making information. By treating partitions coarsely, the model risks discarding temporal dynamics that could provide deeper insights into the neural code underlying behavior.

3. Marginal Improvement Over Baselines: The improvement of the CA-BiLSTM over other methods, such as convolutional neural networks (CNNs), is marginal. This raises questions about the utility of the additional complexity introduced by the CA-BiLSTM framework.

4. Limited Insight into Decision-Making: While the attention mechanism highlights important neurons and brain regions, it remains unclear what new insights this approach provides into decision-making processes. The conclusion that some neurons or regions are more important than others is already well-established in neuroscience.

5. Handling of Spike Train Sparseness: The study highlights the sparseness of spike trains as a challenge in its introduction and title but fails to clearly articulate how this issue is addressed. The partitioning of data, as described, appears equivalent to coarse binning.

Summary

The CA-BiLSTM framework presents an interesting and potentially valuable approach to decoding decision-making from neural spike data, particularly through its integration of an attention mechanism. However, the study’s claims are undermined by unclear advantages over simpler methods, limited utilization of temporal information, and marginal performance improvements. Additionally, the handling of sparse spike data is not adequately addressed. Addressing these issues would significantly strengthen the study’s contributions to the field of neuroscience.

Reviewer #2: Disclaimer: I have formatted the text as markdown. Feel free to render it as such in your preferred editor.

The article proposes a recurrent network architecture endowed with a novel attention mechanism in order to decode mice choice behaviour from spiking neural data across multiple brain areas. The model outperforms other models, both linear and non-linear, and traditional vs ANN (artificial neural network) based, previously used for similar goals. The attention mechanism also allows to identify a posteriori which are the neurons that participate more in the decoding for each mouse, which could be used to identify candidate important circuits necessary for the particular computations that the animals are performing.

## Structural comments

Before pointing out specific comments about the scientific content of the manuscript, I feel obliged to mention a few high level comments regarding its structure:

- The discussion (or conclusions, as the authors write) is very short with just 2 paragraphs, and the first one being a summary of the findings. I think there are many important nuances (as I will get into later) that could have been discussed, but the second paragraph just discusses things without much substance about the paper.

- The supporting information is very lacking. There was only one supplementary figure that I could find, with an *incorrect* caption, which is a copy of Figure 3's (but the figure itself is more like Figure 5 for each mice). There are many details that could have been provided, and which also I will elaborate on further.

From the get-go, these two structural concerns undermine my confidence on the quality of the work that is being presented.

## General comments

The exercise of decoding behaviour from neural activity is not a new one. The novelty in this study is mainly the addition of an attention module to a recurrent architecture (biLSTM) in order to predict choice from spiking data. This has the potential to reveal which neurons are more relevant to decode choice, and in turn establish hypotheses about identifying decision-relevant circuits. However, I believe the authors should provide stronger evidence that this is indeed the case, and that the novelty and value of this idea warrants publication in this journal.

I shall cover now the points that I think that should be considered to this end.

### Model architecture

I would have liked to see some more justification to motivate the architecture of the attention module. In particular, the outputs of the AvgPool and MaxPool operation is fed to an MLP of a certain size. I couldn't even find the details of the size chosen, but why this one? How was this hyperparameter selection done? Also, the pooling operations are indicated to be done 'along the time dimension'. Does this include both time along the trial, and trial number? The output E_c seems to be of size 1xN, where N is the number of neurons (is it? N is not properly defined). I can't find information about the trial dimension. How is the data batched? All this is crucial to then understand the implications of the selection on neurons based on attention weights later on. This part needs to be very clear, and for me at least, at the moment it isn't.

### Data processing

Going now first into trial selection, the 'screening criteria' imposed seem to be quite strict. In the end only 13 sessions of 13 mice survive the process, out of 82 initial mice recording sessions, as per the IBL paper following their exclusion criteria (121 originally). I think it is important that the authors indicate the percentage of sessions that gets discarded in each step, and if it's high they justify the reasons for the thresholds they considered. For instance, how sensitive were the results to the threshold for the minimum number of trials per session? How about the bias criterion? Perhaps using AUC (Area under the ROC curve) instead of classification accuracy to quantify the performance of the model would have allowed to be laxer in terms of class imbalance. Or the data could have been subsampled to balance the classes.

Then, regarding the pre-processing and binning of spike times, I think this is an important step that raises some questions. The choice of the 'segmentation method' is surprising, given the disparity of reaction times (RTs) in the data. The authors show a high correlation between the performance of the chosen segmentation and that of traditional fix window binning of 30ms, with the former slightly outperforming the later. Was the 30ms size the best possible one? It so, it should be stated. The segmentation method leaves me with some doubts, because it assumes some implicit temporal scaling of neural activity, but I don't see why this would be the case. A good sanity check would be to show that, given this segmentation, the model doesn't perform much worse on trials with long reaction times compared with short reaction time ones, which are the predominant type. The whole speed-accuracy tradeoff in the task seems challenging to be reproduced with this selection.

### Task

The IBL task is by now a well stablished experimental paradigm, and it has some special features that are important, but that the authors did not mention. In particular, after the first 90 trials, the trials follow a blocked fashion in which the probability of presentation of the stimulus in one side is higher than the other. There are also different stimuli contrasts, and additional, variability in the RT from trial to trial. Ideally, all of these features could have been taken into account to test in more detail the robustness and performance of the model.

Particularly, rather than just focus on the choice prediction accuracy, it would be interesting to also see what happens when the model misclassifies a trial. Do those classification errors correspond predominantly to difficult trials, or to trials with a particular RT range? Additionally, does the model make the same type of mistakes as the animals, given the stimulus? This would add deeper insight into the value of the model besides a single, global accuracy score that is a few percent higher than that of, for instance, a MLP.

### Attention mechanism

The last third of the manuscript is employed to assess the worth of the proposed attention mechanism in order to determine the impact of different neurons to the mice decisions. The strategy employed consists on ranking the neurons by their attention weight or score, and then training a *different* model architecture without an attention mechanism, in order to show that using the neurons with higher attention weight improves the classification performance as opposed to the same number of randomly selected neurons, or all the neurons. I see this as a valid sanity check, but it is hardly surprising. If the attention model is trained successfully, how could it have been otherwise? Furthermore, I don't understand why the test was performed on a support vector machine, instead of the same BiLSTM architecture just without the attention mechanism. No justification whatsoever is given for this.

The attention mechanism is not the only way to perform feature extraction in ML models. The authors could consider comparing it to simpler methods. For instance, it could be compared with an MLP with L1 regularization in the input weights, which would be a form of context-independent feature selection. After all, the crucial aspect of the attention mechanism is that it's dynamic and adaptive. Hence, the dynamic aspect should be tested and contrasted with a non-dynamic method. Another strategy could be to leverage the task blocks, and to train the model in only one of the block types and test in the other, repeating then the procedure with the opposite type. How does the performance degrade? Are there block specific neurons that are sensitive to the bias? Finally, the attention mechanism could be validated with ground-truth simulated data, perhaps using a Poisson-GLM where one can choose the particular weights of each neuron towards the choice.

Finally, something surprisingly lacking is the combined assessment of the distribution of "important" neurons across brain areas and mice. Are these neurons predominantly located in areas previously identified as decision relevant? All that is shown is distributions for an example mouse from the Witten lab, that doesn't particularly conform to the expectations of distribution of important areas that was primed before as decision relevant, such as LP and PPC (this animal has only 2 PPC neurons, and none of them are selected by the attention mechanism). I believe showing this distribution across mice is crucial to further validate the proposed attention mechanism and position it as a competitive tool to retrospectively identify task relevant neurons.

## Detailed comments

Now, I will refer to more specific comments regarding details in the text.

- Consider replacing words such as "linchpin" and "winnowing" by simpler words (lines 11 and 51).

- Line 60: "preceding decision-making tasks" do you mean "preceding decisions"?

- Lines 61, 78: decision making "direction" - the task hasn't been introduced yet, so "direction" should be avoided here because it's not obvious what it means, rather use "action".

- Lines 72-76: I don't fully get why the "pulse data" is emphasized here, since the spikes are eventually binned. It's not like the raw ephys signal is taken as input.

- Lines 81-83: This is largely repeated, was said already in lines 68-70.

- Line 97: "maintain a time-step memory at time step t". This is confusing, do you mean "a one-time-step memory"?

- Line 119: Consider rewording, performing inference or evidence accumulation is not equivalent to being "constantly contemplating their decision-making"

- Line 121: I disagree with the choice of words "instinctively" and "conscious". For the first, I would rather use "impusively", and the second incorrectly suggests a conscious experience during evidence accumulation.

- Line 122: This needs clarification.

- Line 136: "the form of the algorithm" replace by "algorithmic form".

- Task description: there is no mention of the block structure of the task, the fact that there are different contrasts, or that the task is a reaction time task such that the animals control the duration of the stimulus.

- Table 1: Number of good "nuerons".

- Line 199: Avoid repeating "decision-making times".

- Line 222: "resulting some loss of temporal information" - given that the spike counts are binned in a few segments, is it really the case?

- Fig. 3: "under different segmentation" - missing schemes?

- Line 242: How was the significance assessed? You must report this.

- Fig. 4: Missing y label in a).

- Fig 5: The caption in this figure needs to be clarified. What is the score value? Also it says "nueron" ibl_witten_29

- Line 264: Missing figure?

- Line 266: "consistent with the current research findings" - do you mean your findings? This hasn't been shown.

- Line 282: Again, why the SVM and not a biLSTM?

- Line 290: What exactly is the selection threshold apply onto? This has not been explained.

- Fig. 6: brain "eegion". Also filtering threshold of "0. 1" (extra space). Also the caption should indicate which of the pie charts corresponds to what.

- Fig 7: this is the first time that it's said what "SVM_RBF" means.

**Supplementary information**

The SI should include the following:

- Further details about the model architectures, including the different models shown in Fig. 4.

- Hyper parameter selection.

- Training and validation loss examples.

- Hardware specifications, and model training and evaluation times.

- Details on the data selection criteria, with numbers of trials and sessions excluded at each step.

- A figure with panels like Fig 2b for different bin widths for the the time interval partition, or at least a quantification of the overall performance as a function of bin width.

- A figure like Fig. 6 for each mouse.

References: In general there aren't many references. At least more should be provided when mentioning earlier research on neural correlates of decision making (line 24) and the DDM (line 25). There is not citation for BiLSTMs, or attention mechanisms until late in the text (line 249).

Error bars: in general, the meaning of the error bars in the figures is not specified (figs. 3, 5, 7 and 8).

Presentation: Figures 2 and 3 could be merged. Same goes for figures 7 and 8, or even 6, 7 and 8.

**Have the authors made all data and (if applicable) computational code underlying the findings in their manuscript fully available?**

Reviewer #1: Yes

Reviewer #2: Yes

PLOS authors have the option to publish the peer review history of their article (what does this mean?). If published, this will include your full peer review and any attached files.

Reviewer #1: No

Reviewer #2: No

**Figure resubmission:**
---

## [Decision Letter · Decision Letter 1]

15 May 2025

PCOMPBIOL-D-24-01470R1

Decoding Decision-Making Behavior from Sparse Neural Spiking Activity

PLOS Computational Biology

Dear Dr. Zhang,

Thank you for submitting your manuscript to PLOS Computational Biology. After careful consideration, we feel that it has merit but does not fully meet PLOS Computational Biology's publication criteria as it currently stands. Therefore, we invite you to submit a revised version of the manuscript that addresses the points raised during the review process.

Please submit your revised manuscript within 60 days Jul 15 2025 11:59PM. If you will need more time than this to complete your revisions, please reply to this message or contact the journal office at ploscompbiol@plos.org. Please include the following items when submitting your revised manuscript:

We look forward to receiving your revised manuscript.

Kind regards,

Daniele Marinazzo

Section Editor

PLOS Computational Biology

**Additional Editor Comments :**

This revised version still leaves a lot of space for improvement, in particular some important issues were evaded or addressed unsatisfactorily. In order not to waste your time, and the time of the reviewers, I prefer to give you another chance to carefully and thoughtfully go over these points. Of course, if there's something you don't agree with, you can explain why, and stick to your version, after a proper discussion and rebuttal.

**Journal Requirements:**

 1) We have noticed that you have a list of Supporting Information legends for S1 Table and S2 Table in your manuscript. However, there are no corresponding files uploaded to the submission. Please upload them as separate files with the item type 'Supporting Information'. 2) We have noticed that you have uploaded Supporting Information files, but you have not included a complete list of legends. Please add a full list of legends for your Supporting Information files (Support information.7z ) after the references list. 3) We notice that your supplementary figures are uploaded with the file type 'Figure'. Please amend the file type to 'Supporting Information'. Please ensure that each Supporting Information file has a legend listed in the manuscript after the references list. 4) Please enter the affiliation of Yuhang Zhang in the online submission form. **Reviewers' comments:**

Reviewer's Responses to Questions

Reviewer #1: Authors have addressed my questions

Reviewer #2: ## General comments

I thank the authors for the provided revised materials.

Overall, I do not feel that the authors have addressed with sufficient depth the comments of the reviewers.

I'll refer to each particular comment and what I find lacking about the responses.

### Comment 2.1

Despite the discussion having increased in length, I still find that it could have been elaborated more. There aren't any references or attempts to make meaningful connections with previous literature. There could also have been a more critical stance on the limitations of the work.

The additional supplementary material is appreciated.

### Comment 2.2

The provided additional details helps to clarify most of my doubts. I would still clarify somewhere in the text how the batching is done.

### Comment 2.3

I appreciate the additional information regarding the statistics of the data selection process. It still leaves me somewhat uncomfortable the fact that so many sessions (37) are discarded due to choice imbalance, but on the other hand the authors might argue that they aimed to study close to optimal conditions as a testbed for their method.

Thank you for including the AUC as an additional metric.

Regarding the binning of spike times, Figure 4b raises some doubts. The accuracy curves across almost all methods don't improve as a function of the number of bins, and some of them decrease. It seems hard to justify the choice of 7 bins as optimal. This also introduces the concern that if the results are so robust to the number of bins, perhaps the temporal dynamics of the neural data are not exploited much after all by the methods. This should be clarified.

I still think the implications of the scaling bins at the level of the neural code could have been discussed more.

Regarding the analysis on the RTs, I'm not sure if the authors draw the right conclusions from them. They mention "Interestingly, we found that correctly predicted trials had slightly longer

mouse reaction times compared to incorrectly predicted trials, with an average difference of

11 milliseconds (t-test p-value = 0.031). This suggests a speed-accuracy tradeoff (SAT) that aligns with established neuroscientific principles." While it's true in general there a SAT to be expected, when comparing correct and error trials, the stimulus difficulty should be controlled for. When this is done, there isn't necessarily an expectation for whether errors should be faster than correct responses. On the other hand, the authors also show that correctly predicted trials are associated with easier stimuli (which makes sense), and the fact that they are at the same time slightly slower contradicts the usual SAT (easier trials are faster and more accurate). Overall there seems to be confusion regarding the role of RT in the results.

### Comment 2.4

I appreciate the additional details regarding the task structure and the presence of the blocks.

I could not find, however, where the analysis of the block structure is incorporated in the manuscript, as the authors claim.

Regarding Figure 7 c, since the contrast levels are discrete, a histogram would have been a more appropriate visualization for its distribution.

### Comment 2.5

There isn't still a justification of why the model SVM_RBF is appropriate to perform the analysis in Figure 6. The result should still be introduced more with the character of a validation of the obtained attention weights rather than a result in itself showing evidence for the superiority of the method over others. Probably the same results would have been obtained using one of the other techniques that I'm about to discuss. In fact, since the SVM_RBF doesn't employ the attention mechanism itself, just a selection of the neurons based on the attention weights, appropriate regularization should be able to find a similar selection of neurons if that increases the performance of the model SVM_RBF.

Regarding the new analysis in Figure 7 a and b, I'm missing details about what was exactly done in each case (L1-LogReg, L1-MLP, ANOVA, RFE). There are no references or explanations provided whatsoever. Therefore I'm unable to extract much from this result, despite it showing some evidence in favour of superior performance of the CA-BiLSTM selection.

Finally, Figure 7d shows some appreciated additional information regarding the distribution of "important neurons" across brain regions. However, the discussion about it is confusing. The highlighted regions DG, LP and PPC aren't among the ones with highest weights, they sit rather on the bottom half of the plot (not DG). The authors don't elaborate much on the implications of this result, which could be relevant for the broad Neuroscience community.

### Comments 2.6 and 2.7

Most of these are addressed. I have some additional comments.

The caption of Fig 5 still isn't very clear for me. "Trial select" and "select" are strange labels for the y-axis. The relationship between the three variables shown is still not clearly communicated.

**Have the authors made all data and (if applicable) computational code underlying the findings in their manuscript fully available?**

Reviewer #1: Yes

Reviewer #2: Yes

PLOS authors have the option to publish the peer review history of their article (what does this mean?). If published, this will include your full peer review and any attached files.

Reviewer #1: No

Reviewer #2: No

**Figure resubmission:**
---

## [Decision Letter · Decision Letter 2]

6 Jul 2025

PCOMPBIOL-D-24-01470R2

Decoding Decision-Making Behavior from Sparse Neural Spiking Activity

PLOS Computational Biology

Dear Dr. Zhang,

Thank you for submitting your manuscript to PLOS Computational Biology. 

We are ready to accept your manuscript after you have fixed/clarified a final detail on the normalization of the data reported in figure 7c, see reviewer's comment.

Please submit your revised manuscript within 30 days Sep 05 2025 11:59PM. If you will need more time than this to complete your revisions, please reply to this message or contact the journal office at ploscompbiol@plos.org. Please include the following items when submitting your revised manuscript:

We look forward to receiving your revised manuscript.

Kind regards,

Daniele Marinazzo

Section Editor

PLOS Computational Biology

**Journal Requirements:**

1) Please include the authors' affiliations in the online submission form. Please ensure that the affiliations of the authors listed on the manuscript title page do exactly match with the affiliations provided in the online submission form.

**Reviewers' comments:**

Reviewer's Responses to Questions

Reviewer #2: I congratulate the authors for their effort, which have lead to significant improvements in the manuscript. My comments have been largely addressed.

I don't understand the last panel of Fig 7c (bottom right). How is it normalized? Neither the total correct+error sums to 1 in each bin, nor each conditional distribution of corrercts or erros sums to 1 either. Please revise that detail.

**Have the authors made all data and (if applicable) computational code underlying the findings in their manuscript fully available?**

Reviewer #2: Yes

PLOS authors have the option to publish the peer review history of their article (what does this mean?). If published, this will include your full peer review and any attached files.

Reviewer #2: No

**Figure resubmission:**
---

## [Editor Report · Decision Letter 3]

16 Jul 2025

Dear Dr. Wan,

We are pleased to inform you that your manuscript 'Decoding Decision-Making Behavior from Sparse Neural Spiking Activity' has been provisionally accepted for publication in PLOS Computational Biology.

Best regards,

Daniele Marinazzo

Section Editor

PLOS Computational Biology

---

## [Editor Report · Acceptance letter]

PCOMPBIOL-D-24-01470R3

Decoding Decision-Making Behavior from Sparse Neural Spiking Activity

Dear Dr Wan,

I am pleased to inform you that your manuscript has been formally accepted for publication in PLOS Computational Biology. Your manuscript is now with our production department and you will be notified of the publication date in due course.

With kind regards,

Zsofia Freund
